# ‘As Long as It Comes off as a Cigarette Ad, Not a Civil Rights Message’: Gender, Inequality and the Commercial Determinants of Health

**DOI:** 10.3390/ijerph17217902

**Published:** 2020-10-29

**Authors:** Sarah E. Hill, Sharon Friel

**Affiliations:** 1Global Health Policy Unit, University of Edinburgh, Edinburgh EH8 9LD, UK; 2Menzies Centre for Health Governance, School of Regulation and Global Governance (RegNet), Australian National University, Canberra ACT0200, Australia; Sharon.Friel@anu.edu.au

**Keywords:** gender, inequality, commercial determinants of health, social determinants of health, corporate political activity, tobacco industry, alcohol industry

## Abstract

Scholarship on the commercial determinants of health (CDoH) has sought to understand the multiple ways corporate policies, practices and products affect population health. At the same time, gender is recognised as a key determinant of health and an important axis of health inequalities. To date, there has been limited attention paid to the ways in which the CDoH engage with and impact on gender inequalities and health. This review seeks to address this gap by examining evidence on the practices and strategies of two industries—tobacco and alcohol—and their interaction with gender, with a particular focus on women. We first describe the practices by which these industries engage with women in their marketing and corporate social responsibility activities, reinforcing problematic gender norms and stereotypes that harm women and girls. We then examine how tobacco and alcohol companies contribute to gender inequalities through a range of strategies intended to protect their market freedoms and privileged position in society. By reinforcing gender inequalities at multiple levels, CDoH undermine the health of women and girls and exacerbate global health inequalities.

## 1. Introduction

From where we live, to what we eat, to how we communicate, every aspect of our lives is influenced by the activities of corporations [1]. Recognition of this extraordinary influence—and a desire to document and understand its impacts on health—has led to the emerging field of scholarship known as the commercial determinants of health (CDoH) [2,3,4,5,6,7]. The focus of this field, in both research and practice, has been primarily on the multiple ways in which corporate policies, practices and products influence people’s health [8]. This influence is both subtle and profound, shaping contexts ranging from individual consumption, to urban design, global trade and finance [5].

The relevance of the CDoH is particularly evident in the dramatic rise of non-communicable diseases (NCDs) [4], a major global health challenge that increasingly affects less advantaged population groups [9]. Growth in NCDs is largely driven by the globalised production, marketing and consumption of highly processed foods, drinks and legal drugs, particularly alcohol and tobacco [6]. As noted by a former Director-General of the World Health Organization (WHO), this creates a situation where “efforts to prevent non-communicable diseases go against the business interests of powerful economic operators” [10].

To date, relatively little attention has been paid to the ways in which the CDoH engage with and impact on gender. The concept of ‘gender’ encapsulates socially constructed differences between the sexes, including both gendered roles and behaviours (at the interpersonal level) and the gendered nature of societal norms and organising principles (at the structural level) [11,12]. ‘Gender norms’ refer to prevailing assumptions about the appropriate roles and aspirations of (simplistically) women/girls and men/boys in a particularly society [13,14]. While these norms are most clearly expressed in specific beliefs and behaviours, they reflect the social construction of gender in a particular context [13]. Thus, gender norms are embedded across the multiple layers of society, from formal structures and institutions (at the macro-social level) to specific communities (at the meso-level) to more intimate domestic and individual relationships (at the micro-level).

Prevailing gender norms affect everyone and can damage men’s as well as women’s health [15]. While gendered structural power means men tend to enjoy greater autonomy, authority and economic capital than women, prevailing expectations of ‘masculine’ behaviour mean men are also more likely to engage in risky behaviours - including potentially hazardous use of tobacco, alcohol, motor vehicles and firearms [16]. Gendered expectations and norms are likely to be a significant contributing factor to men’s lower life expectancy [17] as well as women’s higher lifetime morbidity [18].

As a powerful social determinant of health, gender provides a salient lens for examining how the commercial determinants intersect with other social determinants of health. While recognising the fluid nature of gender and its influence on the health of men and those with non-binary identities, we focus here on the CDoH and women’s health. In particular, we examine the ways in which corporations encourage women’s consumption of unhealthy commodities through the creation and reinforcement of gendered norms and stereotypes. We also highlight the ways in which corporations shape institutional and policy settings, which in turn affect women differently than men.

This review draws on secondary research from diverse fields including public health, sociology and anthropology, economics, business and marketing. Where secondary sources are lacking, we also use primary documentary sources—including industry documents and market research. In using the concept of the CDoH, we aim to bring together diverse strands of research to understand how the actions of corporate actors (especially those producing unhealthy commodities) impact on women’s health and on the social, economic and political structures that shape the determinants of women’s health.

There are three sections to this review. The first section describes the practices by which the CDoH seek to influence potential consumers, highlighting how these practices interact with and reinforce gender inequalities in social expectations, roles and aspirations. We illustrate these practices by examining how the tobacco and alcohol industries have engaged with gender in their advertising, corporate social responsibility (CSR) campaigns and sponsorship.

In the second section, we examine the less visible strategies used by the CDoH to maintain their structural power, resulting in the economic and political subordination of women and the exacerbation of gender inequalities in power. Drawing again on the examples of tobacco and alcohol, we illustrate how corporate actors pursue strategic growth in emerging markets, create new markets among female consumers, actively delay the regulation of their activities, and push regulatory limits by tailoring their activities according to political context.

In the third and final section, we reflect on priorities for future research aimed at understanding and addressing the health-damaging impacts of the commercial determinants of health and their impacts on gender inequalities.

## 2. Corporate Practices and Their Implications for Gender Inequalities

Corporations demonstrate a sophisticated awareness of gender in product marketing, which often draws on gendered aspirations and ideals [19]. Such strategies may be particularly relevant for products traditionally associated with masculinity but where companies are seeking to create new markets and demand among women and girls.

At the same time, global businesses are increasingly seeking to portray themselves as socially progressive by engaging with discourses around gender equity in their wider practices, including communication with potential investors and CSR. These discursive and instrumental strategies [20] demonstrate the importance to corporate actors of shaping their perceived role in public and policy debates, ultimately serving to reinforce their structural power as CDoH.

### 2.1. Gender in Marketing—What Women Want?

As anyone who’s watched *Mad Men* knows, engagement with gender norms and stereotypes is a common marketing strategy [19]. Aside from their use to promote health-damaging products, the ubiquity of gendered imagery and messages in advertising can have directly harmful effects by reinforcing problematic, idealised and often unattainable gender stereotypes. Women are more likely than men to be represented as passive, domesticated or “in decorative capacities” [21] while their bodies are more likely to be sexualized [22], visually fragmented or ‘dismembered’ [23]. Such portrayals not only encourage unhealthy aspirations and consumption but also contribute to women’s objectification and disempowerment [21,24].

Gendered messaging is a longstanding feature of both tobacco [22,25] and alcohol marketing [26,27]. Smoking and drinking were historically presented as masculine behaviours, associated with male ‘mastery’ [28]—including mastery of women [29]—while women were often trivialised and/or hypersexualised in advertisements [30,31].

Marketing campaigns targeting women (in high-income countries) started much earlier for cigarettes [25] than for alcohol [26,27]; yet both played on concerns with body image, as in the Lucky Strike “Reach for a Lucky instead of a sweet” campaign [32] and commercials for lower-calorie alcoholic beverages [33]. Companies subsequently sought to link their products with aspirations of sophistication, independence and attractiveness [26,32], deliberately displacing traditional connotations of women’s smoking and drinking as “louche and libidinous behaviour” [25] unsuited to the feminine ideal. Recognising the significance of women as a potential market [34], tobacco and alcohol producers developed products targeted at female consumers—including slim cigarettes and ‘feminine’ packs [32,35,36], ready-to-drinks alcoholic beverages [37], sparkling wines [38], and—in the US market—hard seltzer or soda drinks [39].

Female-targeted marketing has evolved in line with changing social norms. In the 1980s, for example, both tobacco and alcohol advertising referenced the women’s liberation movement by associating their products with aspirations of female emancipation and realisation. Such referencing can be seen in Philip Morris’s “You’ve Come a Long Way, Baby” campaign [35] (Figure 1) and consonant advertisements presenting women’s alcohol consumption as a marker of their independence [26,27]. Corporations thus “recirculat[e] traditional gender messages in new ways” [21], appearing to celebrate female empowerment while actually subverting it to associate a product with its fulfilment.

The cynicism underpinning these practices is evident in internal industry discussions around placement of the Virginia Slims ad in an African American-targeted magazine:


*Frankly, we weren’t sure, with our theme “You’ve come a long way, Baby”—that we could run this advertising in Ebony, but why not? As long as it still comes off as a cigarette ad, not a civil rights message.*
(Hal Weinstein, 1969 [42].)

As this statement suggests, unhealthy commodity industries are willing to associate their brands with discourses of empowerment so long as these align with prevailing social norms and do not actually challenge dominant systems of privilege and subordination. In relation to women, Törrönen & Rolando link such representations “with a neo-liberal message about [female] autonomy, expressed appropriately through consumer choices” [26]. In other words, women are encouraged to express their freedom and individuality through their consumption—although they continue to be represented as a fashionable, slim, and a sexually-attractive ideal. Despite its invocation of female emancipation, Törrönen & Rolando note that such advertising “does not empower women as multi-dimensional whole subjects” but rather “repeat[s] ‘progressive’ gender representations while reinforcing traditional gender norms and expectations” [26].

This corporate cynicism is also evident in companies’ responses to public health concerns and the associated advertising restrictions. Historically, tobacco companies targeted women (who were considered more health-conscious) with low-tar and ‘light’ cigarette brands in response to growing public anxieties about the health risks of smoking [32,43]. As advertising bans were introduced in high-income settings [44], cigarette promotion moved to sports sponsorship (baseball [45] and motor racing [46,47] for men, tennis [48] and fashion [32] for women). Sports sponsorship has similarly become a forum for the promotion of alcohol brands: Budweiser (owned by AB InBev) has for many years sponsored the US Women’s National Soccer Team, and recently became a key sponsor of England’s Women’s Football team [49]. Diageo—the world’s leading spirits producer—supports women’s cricket in India [50], one of their most important markets [51].

Manipulation of gendered ideals and aspirations in their advertising has paid significant dividends for tobacco and alcohol companies. The targeting of women in product design and marketing helped normalise female smoking and drinking in high-income countries [25,52], contributing to women’s equal representation with men in advanced cigarette and alcohol markets [32,53]. As markets in high-income countries have ‘matured’, companies have shifted their focus on female consumers to emerging markets [54,55]—as illustrated by comments from a company director in India:


*The rise of women consumers offers a great opportunity for us to grow in the future. That is a target segment we need to keep in mind and ensure that our brands get more bilingual and speak to both sets of audiences, not just be male-centric.*
(Bhavesh Somaya, Diageo India [56].)

### 2.2. Gender in CSR—Empowering Women?

As pressure mounts for governments to restrict the advertising of unhealthy commodities, companies increasingly seek to promote their public image and political influence via CSR activities and donations to charitable organisations [57,58,59,60]. Women’s ‘causes’ have long been a target for such practices, particularly by tobacco companies [48,61] which—by the late 1980s—were contributing an estimated US$4.5 million to minority and women’s organisations in the US alone [62]. The potential for such contributions to deflect attention from the industry’s harmful impacts is illustrated by a quote from the former director of the Women’s Research and Education Fund, who—when asked about the organisation’s receipt of over USD 100,000 from tobacco companies—admitted “to tell the truth, I’m not that interested [in the health concerns]. I’m just glad they fund us” [63].

‘Women’s issues’ provide an avenue for tobacco and alcohol companies’ efforts to present themselves as responsible corporate actors [61,64], although research in this area is limited (particularly in relation to alcohol sponsorship). International Women’s Day has become a point of focus for both industries. British American Tobacco (BAT) has received awards for its International Women’s Day campaigns [65] and commendation for how “the development of BAT products provides a livelihood for many women” [66]. Several alcohol companies have released limited-edition products to ‘support’ International Women’s Day [67]—including craft beer company BrewDog’s Pink IPA [68] and Diageo’s ‘Jane Walker’, a repackaged version of their Black Label Johnnie Walker [69] (Figure 2). The latter was originally designed for release following the 2016 US presidential election on the assumption that Hillary Clinton would win [70], but its launch was subsequently delayed and reframed as a celebration of International Women’s Day.

Both industries are increasingly focusing their CSR activities on female ‘empowerment’ in low- and middle-income countries, where women represent a growing market [32]. Philip Morris International (another cigarette giant) sponsors women’s empowerment programmes in China and self-help groups in Argentina [72]. Spirits producer Diageo partnered with an international charity to support programmes combating gender-based violence [73,74] and a petition to ban violence and harassment in the workplace [75]. AB InBev—the world’s largest beer company—took a high-profile role in South Africa’s #NoExcuse campaign against gender-based violence [76] with their Carling Black Label beer featuring prominently in campaign materials [77] (Figure 3). Initiatives such as these serve to create a ‘responsible’ public profile for alcohol companies while simultaneously promoting their reputation and brands [78] and distracting from the role of alcohol in violence against women [79].

### 2.3. Summary: Corporate Practices and Gender Inequalities

Unhealthy commodity industries exploit gendered ideals to promote their products. Tobacco and alcohol companies have responded to changing gender norms by appearing to align themselves with female empowerment and the promotion of women’s causes. Such promotions have helped create new markets among female consumers—thus increasing profits—while portraying these industries as progressive, supporting their image as important social partners and reducing the likelihood of regulatory controls on their activities. At the same time, these practices associate women’s emancipation with consumption of potentially addictive substances while reinforcing gendered expectations and stereotypes.

## 3. Corporate Strategies, Structural Power and Gender Inequalities

Marketing and CSR practices provide the most obvious illustration of corporate engagement with gender. However, sitting behind these practices are strategies that reinforce the structural and economic power of the commercial determinants of health. These strategies create the conditions that exacerbate gender inequities, undermining various social determinants of health for girls and women.

Corporations structure gender inequities via their strategic engagement with markets, international trade, and domestic policy. These strategies act to protect companies’ market freedoms and economic influence. This aspect of their interaction with gender is largely invisible, since it does not involve the overtly different treatment of women and men. Instead, it relies on structures and processes that reinforce the dominant position of powerful corporations, which indirectly disadvantage women and girls in terms of economic and political power.

Feminist critiques of capitalism emphasise the extent to which the market undervalues work traditionally undertaken by women and girls—including caring, domestic (re)production and early education—while commodifying more typically ‘masculine’ activities [12,81]. This distinction became formalised in the nineteenth century when the creation of the ‘corporation’ effectively consolidated the subordination of non-market to market work [12]. The discounting of female endeavour was reinforced through the subsequent privileging of free market goals in development of social and legal systems, producing a ‘market society’ [82] in which the kinds of work traditionally undertaken by women receive little or no remuneration [81].

The extension of feminist critiques to economic globalisation highlights how macro-structural models reproduce gender biases [12]—a process described by Beneria as “‘economic man’ gone global” [81]. Global changes in the labour market mean women are now overrepresented in low-paid and casual employment while simultaneously bearing primary responsibility for unpaid household and caring work [83]. At the same time, globalisation has resulted in a “narrowing of national policy space” [83] and reduced public spending in health, education and social care—again, with disproportionate impacts on women.

The resulting gender inequalities are staggering. Women globally are over-represented in vulnerable and informal employment, more likely to be unemployed, and—when they are employed—earn almost a quarter less than men [84]. At the same time, the value of women’s unpaid work is estimated at between 10 and 39% of countries’ GDP [85].

In addition to women’s economic disadvantage, the dominance of men in positions of power reinforces a more aggressive and confrontational culture in many boardrooms, parliaments and workplaces, resulting in the systemic privileging of ‘masculine’ aspirations and interactions. *The Economist* estimates that women comprise just 7% of government leaders, 15% of company board members and 3% of chief executives, and that—based on current trends—it would take over 200 years for this gap to close [86]. The systemic marginalisation of women means their concerns and causes are less likely to be prioritised by those in authority, perpetuating a structurally-embedded gender bias that contributes to gender inequalities in the social determinants of health.

This brief critique of capitalism and economic globalisation highlights the role of the commercial determinants of health as key architects and actors within global market systems, exerting enormous influence on the gendered nature of economic power. In addition, consistent with broader analyses of corporate influence on public policy [87], the structural power of the CDoH is often implicit and institutionalised. Thus, gender inequities are reproduced and reinforced by the CDoH without overt action or discrimination on the part of the relevant actors, since the prevailing neoliberal economic paradigm is inherently gendered [12]. In other words, the strategic actions of powerful corporate entities (including multinational tobacco and alcohol companies) reinforce a prevailing philosophy that privileges economic objectives and market freedom over other social goals, thus buttressing structural inequalities between women and men.

As outlined below, industries such as tobacco and alcohol play an active role in maintaining this market-based social paradigm and the resulting gender inequalities. Key corporate strategies include the consolidation of these industries and their expansion into emerging markets; the deliberate creation of new markets among women and girls; tactics used to prevent or delay regulation; and exploitation of regulatory gaps or deficiencies, particularly in low- and middle-income settings. These strategies allow corporations to maintain their dominant position in the social and economic landscape, reinforcing the privileging of market power and the subordination of the non-market aspirations and endeavours that form the core of many women’s lives.

### 3.1. Industry Consolidation and Expansion Into Emerging Markets

Leading tobacco and alcohol companies have actively sought to expand their markets while consolidating ownership through acquisitions and mergers. This is particularly pronounced in the case of tobacco, where the twentieth century saw many smaller companies and former state monopolies come under the control of larger producers—such that two thirds of all cigarettes are now sold by just three multinational companies [88]. While the alcohol market is less consolidated, it is still a highly concentrated industry with a handful of companies dominating most markets [59,89,90].

In recent decades, these major companies have actively sought to expand their markets in low- and middle-income countries. From the 1980s onwards, with cigarette sales declining in high-income countries [88,91], tobacco companies dramatically increased their investment in low- and middle-income countries [92]. Similarly, alcohol companies have increasingly sought to expand their markets in Asia [54,93], Latin America [94], and sub-Saharan Africa [90,95].

Both tobacco and alcohol companies lobby governments and use legal action to remove trade barriers [95,96,97], while tobacco companies have also used smuggling to establish their brands in new markets [43]. Market liberalization [98] and industry expansion [59,90,92] have led to growing tobacco and alcohol consumption in Latin America, Asia, Eastern Europe, Africa and the Middle East [88,89,90,92,95,99,100,101].

The profits of global trade in alcohol and tobacco benefit corporate investors rather than local populations. The headquarters of the major tobacco and alcohol companies are located in high-income countries [90,101], yet the majority of production and consumption occurs in low- and middle-income countries [88,89,90,99,102]. The economic value of these companies to their host countries ensures the political support of governments such as the UK and the USA in efforts to remove regulatory barriers from target countries [95,97,103]. This structural corporate power [87] makes it difficult for these governments to take concrete steps to protect global health and gender equity where such measures are in tension with corporate interests.

### 3.2. Companies View Women’s Increasing Economic Independence as an Opportunity to Expand Their Markets

Women represent an important area of market growth for unhealthy commodity industries. As women gain greater labour market representation and economic independence, they attract interest from companies as potential consumers of tobacco, alcohol and other products [43,104,105,106]. In 1928, the President of American Tobacco compared the recruitment of women as smokers with “opening a new gold mine right in our front yard” [34]. The same logic underpins alcohol companies’ efforts to recruit women consumers in low- and middle-income countries [38,107]. A recent analysis of the alcohol market in the Asia Pacific region notes that “[m]ore women in the workforce and rising household incomes aid value growth … [the] shift … towards a female-inclusive, more gender-equal workforce, is expected to contribute to wine consumption” [55].

Growing economic independence, alongside targeted marketing, is likely to lead to higher future tobacco and alcohol consumption among women in low- and middle-income countries. Modelling based on historical patterns of cigarette use suggests that—over time—female smoking in high-income countries followed a similar pattern to that seen in men, though with a later peak [108]. There is also a clear association between gender equity and women’s tobacco use at a country level [109], suggesting a link between economic and social independence and recreational consumption. Alcohol use is likely to follow a similar pattern, as suggested by forecasting which indicates female drinking will more closely resemble that of men in future decades [110].

### 3.3. Companies Actively Resist Market Regulation

Both tobacco and alcohol companies actively resist market regulation through a range of strategies [111] that are particularly well documented in the case of tobacco [43,112,113,114,115]. One such strategy is industry self-regulation, where companies pre-empt legislation by introducing their own voluntary codes of conduct. Cigarette producers first introduced voluntary marketing codes in the 1960s [116], and while evidence suggests these are largely ineffective [102,117] they are still promoted in many low- and middle-income countries [43]. Alcohol companies use similar codes, and—unlike tobacco—have largely avoided more stringent advertising restrictions. Voluntary marketing codes are ineffective in preventing marketing strategies that target young people or present alcohol as enhancing femininity or masculinity [118] but are effectively used by companies to argue against the need for regulation [119,120]. In Brazil, for example, self-regulatory codes have allowed the alcohol industry to carve out exemptions from legal advertising restrictions for many beers and wines [94].

Alcohol producers have taken self-regulation a step further via the development of ‘social aspect organisations’: industry-funded organisations that generate public education campaigns, such as ‘DrinkAware’ in the UK [121] the ‘Alcohol101′ campaign in the US [122], and various ‘responsible drinking’ campaigns in emerging markets (such as Africa) [123]. Such campaigns are largely ineffective in changing drinking behaviour but are frequently presented as an ‘alternative’ to measures such as compulsory health warnings [90,120,121]. Industry-funded campaigns invariably emphasise individual responsibility in avoiding alcohol-related harm [121]. In addition to being largely ineffective, there is indirect evidence that campaigns focused on ‘binge’ or unsafe drinking may reinforce gendered stereotypes by disproportionately presenting young women as ‘guilty’ of such behaviours (Figure 4) [124].

### 3.4. Companies Use Tactics in Low- and Middle-Income Settings That Are Restricted in High-Income Countries

Alcohol and tobacco companies exploit the less stringent regulatory context of many low- and middle-income countries, using aggressive marketing and lobbying strategies that are illegal or significantly curtailed in high-income countries [43,120,126]. Tobacco companies have been particularly aggressive in this regard, exploiting contexts where Ministries of Health lack sufficient power to introduce or enforce legislation and limited public resources make governments more vulnerable to industry intimidation [43,92]. In Kenya and Uganda, for example, BAT has used legal challenges to block tobacco control legislation [127], while elsewhere in the region, employees have reportedly bribed government officials to act in the company’s interests [128]. In the words of whistleblower Paul Hopkins, company managers consider such tactics “the cost of doing business” in Africa [129].

Where there are advertising restrictions, the implementation of these is often weaker in low- and middle-income countries [43,120,130]. This has real implications for population exposure to promotional campaigns. The number of tobacco advertisements is estimated to be over 80 times higher for communities in low- and middle-income settings compared with those in high-income countries [131]. In relation to alcohol—where advertising restrictions are less advanced than for tobacco—companies often use tactics in emerging markets that would be problematic in high-income countries. In Africa, for example, companies such as Diageo have sponsored beauty pageants including Miss World Kenya [123], while alcohol advertisements appear frequently in women’s magazines [132]. In the Philippines, billboards advertising a premium brandy used the strapline “Have you tasted a 15 year-old?” When the campaign was challenged following complaints from local women’s groups, the manufacturer defended the campaign by claiming no sexual innuendo was intended [133].

In addition to reinforcing highly problematic gender stereotypes, these tactics allow corporate actors to expand their markets in low- and middle-income countries, thus increasing exposure to unhealthy commodities (particularly among women), reinforcing market dominance, and exacerbating gender inequalities in economic and political power.

### 3.5. The Impact of Industry Strategies on Women’s Health and Gender Inequalities

We argue that—taken together—the above strategies can be expected to have a detrimental impact on women’s health and on gender inequalities in the underlying determinants of health. Evidence indicates the expansion of multinational cigarette and alcohol companies into emerging markets has been particularly detrimental for women’s and girls’ exposure to health-damaging products. Women in these settings have traditionally had very low levels of tobacco and alcohol use [30,54,93,94,134]. The opening of formerly-protected domestic markets to foreign companies has produced exponential growth in female consumption of cigarettes and alcohol—as in South Korea, where smoking among young women increased from 1.6% to 13% in just 10 years [135]; and India, where alcohol consumption has increased dramatically since the 1990s [54] with women estimated to account for 25% of this growth [136]. The impact of trade liberalisation is particularly evident in the countries of the former Soviet Union, where smoking among women doubled in the decade following the entry of multinational tobacco companies to local cigarette markets [137].

Consistent with the challenge of demonstrating causality in relation to the social determinants of health [138], it is harder to find direct evidence of how specific industries impact gender inequalities. Here, our analysis relies on broader evidence concerning the impacts of economic globalisation on gender inequalities in income, status and influence [81,83,84,86], and on critical theory—particularly feminist critiques of economic globalization and development [12,81,139]. Such theory argues that multinational corporations play an active role in maintaining gender inequalities by reinforcing the privileging of market goals in the organisation and regulation of society. Thus social, political and legal structures are dominated by market-based norms that systematically under-value women’s (re)productive activity and disadvantage women and girls in terms of economic and political power.

### 3.6. Summary: Corporate Strategies and Gender Inequalities

While corporations such as tobacco and alcohol engage explicitly with gender in their marketing and CSR practices, the strategies via which they maintain their structural power [87] are largely invisible. The effectiveness of these strategies is reinforced by a predominantly neoliberal understanding of social dynamics, which interprets freedoms and responsibilities primarily in terms of individual actions [140]. This allows corporations to highlight their more visible practices (such as corporate sponsorship and employment policies) while deflecting attention from the extent to which their core activities reinforce gender inequities at a structural level.

Key strategies via which corporations reinforce their structural power include the active expansion of their markets (including the recruitment of new female consumers) and tactics for resisting state regulation. These comprise voluntary marketing codes, campaigns led by social aspect organisations, and the exploitation of weak state regulation via dubious lobbying and marketing strategies, particularly in low- and middle-income settings. Such strategies allow corporations to maintain their privileged position in domestic and international systems of trade and commerce, thus reinforcing the inherently gender-biased system of globalised capitalism.

This plays out in various ways. Unregulated global markets increase economic inequalities, pushing women into more vulnerable forms of employment while they still bear primary responsibility for unpaid ‘domestic’ labour [83]. The privileging of economic growth via market expansion reduces the public protection of health, social care and education [81,83]. Ultimately, these biases undermine the social determinants of health for women and girls: “Girls, in some contexts, are fed less, educated less, and more physically restricted; women are typically employed and segregated in lower-paid, less secure, and ‘informal’ occupations” [83].

As noted by the Women and Gender Equity Knowledge Network of the WHO Commission on the Social Determinants of Health (CSDH) [83], “[g]ender relations of power constitute the root causes of gender inequality and are among the most influential of the social determinants of health” [141]. The CDoH reinforce and exacerbate unequal gender relations of power through the active promotion and protection of their market and social domination.

## 4. Conclusions

This review has examined the relationship between gender and the commercial determinants of health, demonstrating how the strategies and practices of corporate actors interact with the social determinants of health in ways that exacerbate gender inequities and exploit gendered social norms and relationships. Using the tobacco and alcohol industries as examples, we illustrated how large corporations exploit gender norms and stereotypes in their marketing; present themselves as socially progressive actors through their public engagement with women’s causes; and exacerbate gender inequalities in power by reinforcing their structural dominance and the privileging of the free market.

At the most fundamental level, these activities reinforce unequal gender relations of power by perpetrating the dominance of the market in both domestic and global spheres, reinforcing societies’ focus on economic growth to the detriment of other social goals. Women and girls are multiply disadvantaged by this commercial bias, which systematically undervalues labour undertaken in the domestic or ‘private’ sphere while simultaneously reducing governments’ capacity to redistribute wealth within their populations and ensure the provision of essential services such as education, social care and primary and reproductive health care [83]. Thus, the actions of these companies reinforce women and girls’ disadvantage in relation to the social determinants of health.

At the meso- and micro-levels, we again see the gendered impact of corporate activities in terms of the physical, social and cultural environments in which women and men live their lives. Multinational businesses exert a profound influence on our everyday surroundings in ways that not only promote unhealthy consumption [5] but also create unhealthy expectations. Health damaging industries manipulate prevailing gender norms in advertising their products to women and men, often in ways that reinforced problematic gender stereotypes.

In summary, we argue that the CDoH interact with gender at the micro-, meso- and macro-levels in ways that damage women’s health, reinforce harmful gender norms and expectations, and perpetuate the structural drivers of gender inequalities. These impacts reflect corporate engagement in marketing and CSR, in shaping policy environments, and in the broader political economy of trade liberalization and economic globalization. While further evidence is needed to understand and unpack these interactions, the concept of the commercial determinants of health is helpful in synthesizing evidence and insights from diverse literatures to examine the relationship between corporations, women’s health and gender inequalities.

### Limitations of the Current Evidence Base, and Priorities for Future Research

Our analysis is a starting point, often posing more questions than answers. More research is needed to unpack the pathways between the CDoH and gender inequalities in health, and to inform efforts to address these inequalities. Such a research agenda could address three questions: (1). How do corporate activities engage with and impact on gender/gender inequalities in health? (2). What are the mechanisms by which multinational corporations influence the determinants of health for girls and women? and (3). How might policy makers, advocates and communities mitigate the impact of corporate activities on gender inequalities?

While it is beyond the scope of this review to set out a comprehensive plan for future research, we highlight below some specific areas where further empirical enquiry would be valuable. The majority of these relate to the first of three research questions described above, which aligns with the focus of this review. However, we are also conscious of the need to move beyond describing these relationships to explore potential points and means of intervention that might reduce the negative impacts of the commercial determinants of health on gender and gender inequalities.

This paper has focused on the activities of the tobacco industry, where a small number of health researchers have examined companies’ engagement with gender, and the alcohol industry, where most of the evidence is drawn from other literatures (e.g., marketing journals) or industry publications. There is even less health-focused research examining the significance of gender for producers of ultra-processed foods and beverages—an industry that has engaged with gendered expectations and stereotypes for as long as it has been advertising [142], and which is known to employ similar corporate strategies to those of tobacco companies [6,143,144]. Similar practices and strategies are used by other large-scale businesses including the gambling industry [145], the automobile and fossil fuel industries [5], and private healthcare companies [146]. There is currently a dearth of health research examining how these industries engage with gender or analysing their impacts from a gendered perspective. In outlining how the CDoH engage with gender, this paper has drawn extensively on research undertaken in high-income settings—which reflects the much more limited availability of evidence from low- and middle-income countries. However, our analysis highlights the extent to which corporations often use more extreme tactics in contexts with limited regulatory capacity, weaker mechanisms of public accountability and/or more limited research capacity. There is an urgent need for more research to uncover the practices and strategies of the CDoH in resource-limited settings, particularly in terms of their exploitation of gender stereotypes and their impacts on gender equity.

This analysis has primarily focused on the relationship between the CDoH and women—but men are also negatively impacted by the activities of these industries and bear a disproportionate burden in relation to the impacts of tobacco consumption [134] and many other self-harming behaviours. Further research is needed to understand how the CDoH exploit stereotypes and expectations of masculinity in marketing their products, and how their campaigns influence prevailing gender norms. Research is also needed to understand how the dominance of large corporate actors shapes structural gender dynamics in ways that undermine the capabilities of men as well as women. While this is a relatively neglected area in health scholarship [147], advocacy organisations such as *Promundo* [148] have highlighted the impacts of harmful masculine norms and the need for gender-transformative approaches.

We also found limited evidence exploring the impact of specific industries on gender inequalities in earnings, workers’ conditions, political empowerment and other social determinants of health. Further research is needed to understand how the entry of multinational industries into local markets impacts the social determinants of health for women and girls, and what measures might be taken to protect health and gender equity in this context.

The addictive nature of nicotine means the tobacco industry is relatively resilient to economic recession, with business analysts predicting it will do better than most in the wake of COVID-19 [149]. Other health-damaging industries similarly benefit from the embedded nature of their products’ consumption [150], making them highly resilient to economic downturns. Whatever other challenges we can anticipate in future decades, the commercial determinants of health will continue to exert a powerful influence on the health of women and men.

In 2001, the US Surgeon General issued a call to “[e]xpose and counter the tobacco industry’s deliberate targeting of women and decry its efforts to link smoking … with women’s rights and progress in society” [151]. This same call is relevant to other health-damaging industries, where wider knowledge of their strategies and tactics will engender greater public support for regulation to protect the health of women and other groups targeted by their activities.

As Beneria [81] argues, we can choose to recalibrate the role of the corporation in our societies and the extent to which market freedom is privileged over other collective aspirations. The commercial determinants of health present a major challenge for governments in seeking to protect and promote the health of women, men and all citizens. They deserve our scrutiny.

## Figures and Tables

**Figure 1 ijerph-17-07902-f001:**
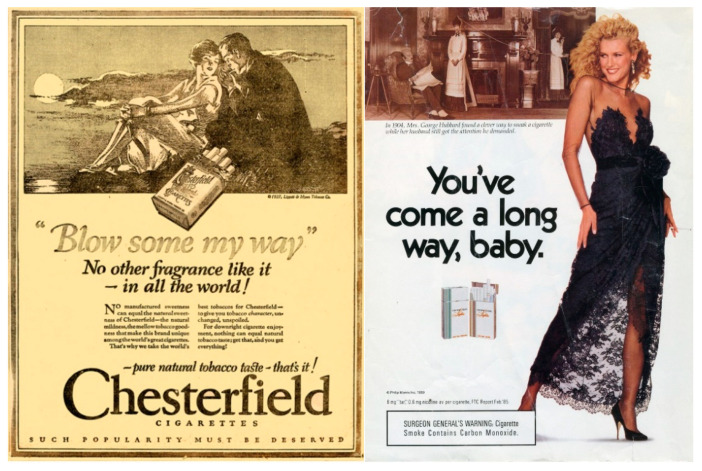
Women and cigarettes—advertisements from 1927 (Ligget & Myers Tobacco Company) [40] and 1989 (Philip Morris) [41].

**Figure 2 ijerph-17-07902-f002:**
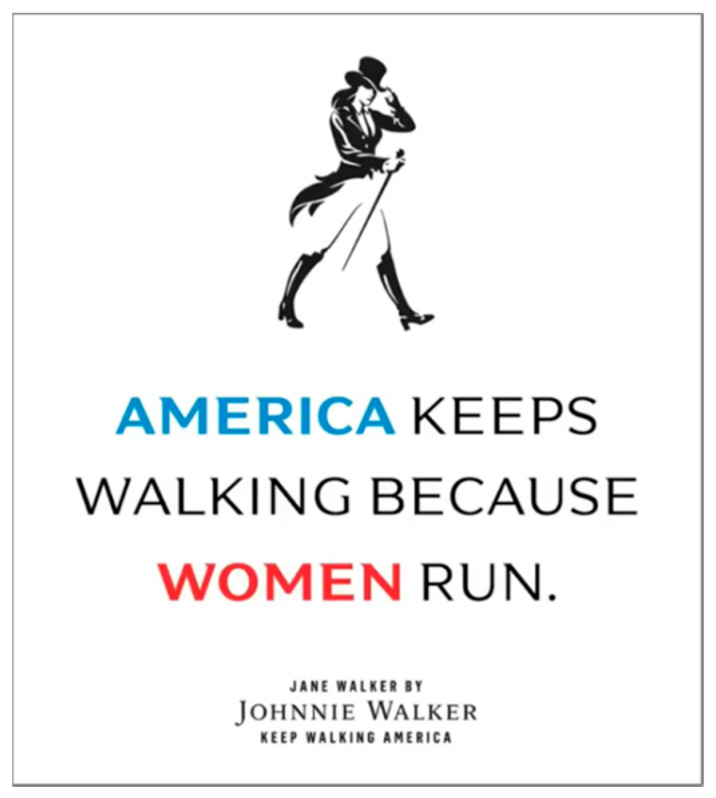
Image from ‘Jane Walker by Johnnie Walker’ promotional video [71].

**Figure 3 ijerph-17-07902-f003:**
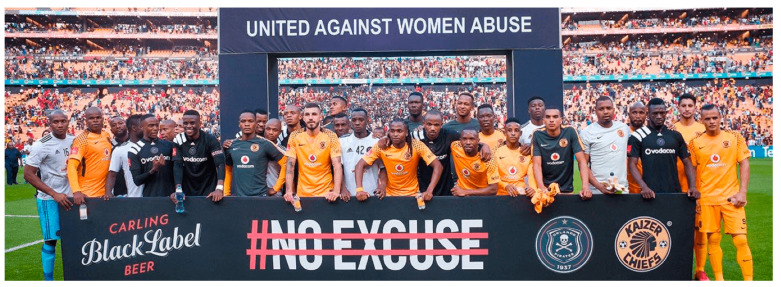
#NoExcuse campaign by Carling Black Label, South Africa [80].

**Figure 4 ijerph-17-07902-f004:**
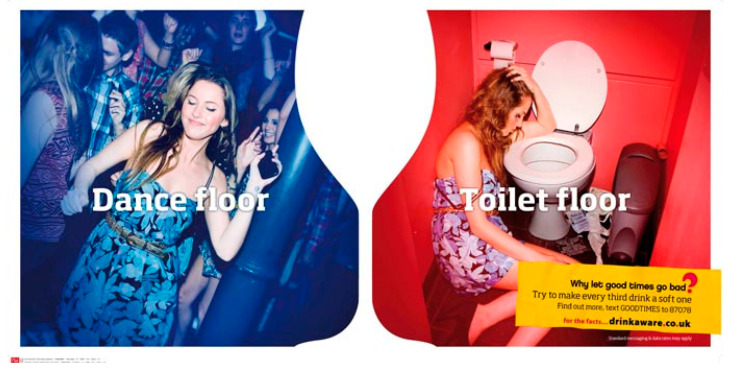
DrinkAware campaign for responsible drinking [125].

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
