# Peer review of "‘As Long as It Comes off as a Cigarette Ad, Not a Civil Rights Message’: Gender, Inequality and the Commercial Determinants of Health"

_ijerph, 2020, doi:10.3390/ijerph17217902_

Round 1
Reviewer 1 Report
Dear Editor
This paper is a very good contribution to the field and it is well written and well organized.
I only suggest that, in the conclusion, the authors tighten up more the main arguments of the paper: that the tobacco and alcohol corporations do more harm than good to women's health, consolidate traditional gender norms and relations and perpetuate gender inequalities.
I also suggest that the last section "Limitations of the current evidence base, and priorities for future research" be titled as follows:
Conclusion
The paper is publishable.
Reviewer 2 Report
There is a feeling that the authors are some biased as a result of reading the proposed manuscript. The exploitation of gender equality by business can be just as significant as the exploitation of the "preferences" of a male or female subculture (for example, the body). In this sense, the article has some a political message. It is necessary to distinguish between research on the exploitation of social topics (including women's rights) by business and the promotion of consumption behavior and marketing moves... However, this perspective on the problem of commercial determinants of women's or men's health can be acceptable in the scientific literature. It would be appropriate at the end of the manuscript to draw scientific conclusions about further steps in the study of this problem, and outline ways of empirical research.
Reviewer 3 Report
This is an interesting review paper discussing the impacts of tobacco and alcohol companies' commercial determinants of health on women and girls. There is a lot of rich information synthesized from previous studies, which makes it an easy and informative read.
In general, I am sympathetic to the authors' argument and goal in the paper. However, while many assertions about the impacts of these industries on the health of women and girls, the paper is missing the crucial empirical links that show precisely how women and girls are impacted. We are given no statistical data on any measurements of how women and girls are impacted, such as longitudinal data on how the entry of these industries into local markets have directly contributed to depressed wages for women, rising income inequalities for women, restricted political access for women, higher rates of domestic violence against women, higher rates of illness/disease in women/girls directly related either to consumption of the products or the side-effects of work-related hazards, etc. (especially in developing nations, which the authors mostly focus on).
Since it is a review paper, the authors need to rely on secondary sources--but there must be some sources that the authors could find to bolster the empirical part of their analysis. Without this, we are left with a manuscript which comes across as a rather lopsided literature review (in the sense that we are not presented with literature that gives us the empirical data to support the assertions). In short, the paper has promise in its skeletal framing of the issue, but requires some empirical flesh to give it real form. If it were to have more empirical data, it would then be publishable as a review paper.
Additional issues regarding specific passages are outlined below:
Lines 20-21: "the practices by which the CDoH engage with women in their marketing and corporate social responsibility activities"
--> This sentence is confusing;the relative pronoun "their" makes it sound like CDoH has marketing and CSR activities, but of course companies engage in marketing and CSR activities which produce CDoH-related effects, so it should be rewritten to emphasize that you are talking about companies here.
Lines 23-24: "how the CDoH contribute to gender inequalities through a range of strategies intended to protect their market freedoms and privileged position in society."
--> Likewise, the relative pronoun "their" is again seeming to say that CDoH are trying to protect their market freedoms and privileged position in society, not companies. Needs to be rewritten.
Line 69: differentially to men --> "differently than men"
Lines 212-213: "strategies that aim to create and reinforce the structural and economic power of the commercial determinants of health"
--> I find it hard to believe that alcohol and tobacco companies are crafting strategies which are AIMED at creating/reinforcing anything related to CDoH. Rater, CDoH seem to be the byproducts (largely unintended, I would presume--I can't imagine that these companies in fact WANT to create disease and illness in their customers). So I would suggest rephrasing this sentence.
Lines 249-255: The invocation of neoliberalism in "neoliberal globalization" and "neoliberal economic paradigm" does not seem to add much, and the concept of neoliberalism is not really interrogated, so I suggest removing this kind of reference. Conversely, if the authors can more clearly connect neoliberalism with gender and CDoH practices, and highlight what new effects neoliberalism has had, then this would be a welcome addition to the paper. (For example, gendered advertisement and marketing for cigarettes, alcohol, and many other products can be found dating back to the early 1900s in many countries, and they were unrelated to neoliberalism.)
Lines 377-383: Here the authors are trying to link market expansion with negative health impact on women and girls, but the empirical link is missing. The authors state this connection (causation) several times throughout the text, but it comes across as repeated assertions, without the data to make this explicit. The few anecdotes about potentially sexist advertising do not make the broader argument clear; empirical evidence is needed. For example, do these companies actually pay women less than men, or hire women only part time? Or is there scientific data tracking the increase in alcohol consumption in these "emerging markets" with specific health effects? Or better yet, could the authors draw more clearly from literature that has empirical data, such as interviews with women and girls who live in these countries and have been impacted by these companies and their products?
Lines 398-405: Again, we get assertions about the impact of alcohol & tobacco company-related CDoH on women and girls and on the private/domestic sphere, but we still do not have specific examples or empirical data related to this. The authors should find some way to support these assertions with some kind of measurements or data. While I agree with the authors that there is likely this kind of impact, it is incumbent upon us to prove that impact.
Round 2
Reviewer 3 Report
The revised manuscript shows sufficient effort in addressing my initial concerns regarding the weaknesses of the paper, and I appreciate the authors' responsiveness and willingness to address these issues. While the design of paper cannot be significantly improved (e.g., it is impossible to add primary data to complement the secondary data-based assertions), the authors have satisfactorily framed the purpose and limitations of the present study, and thus I feel that it is ready for publication.